



# An intercomparison of aircraft sulfur dioxide measurements in clean and polluted marine environments

Loren G. Temple[1], Stuart Young[1], Thomas Bannan[2], Stephanie Batten[3], Stéphane Bauguitte[3], Hugh Coe[2], Eve Grant[1], Stuart Lacy[1], James Lee[1], Emily Matthews[2], Dominika Pasternak[1], Samuel D. A. Rogers[1], Andrew Rollins[4], Jake Vallow[1], Mingxi Yang[5], Pete M. Edwards[1]

[1]Wolfson Atmospheric Chemistry Laboratories, Department of Chemistry, University of York, York, YO10 5DD, UK
[2]Department of Earth and Environmental Sciences, University of Manchester, Manchester, M13 9PL, UK
[3]Facility for Airborne Atmospheric Measurements, Cranfield University, Cranfield, MK43 0AL, UK
[4]Chemical Sciences Laboratory, National Oceanic and Atmospheric Administration, Boulder, CO, USA
[5]Plymouth Marine Laboratory, Prospect Place, Plymouth, PL1 3DH, UK

*Correspondence to*: Pete M. Edwards (pete.edwards@york.ac.uk)

**Abstract.**

The University of York's laser-induced fluorescence (LIF) instrument for measuring sulfur dioxide ($SO_2$) was compared to a commercial pulsed fluorescence (PF) and iodide chemical ionisation mass spectrometer (I⁻CIMS) aboard the UK FAAM research aircraft in both remote and ship-polluted marine environments. In high $SO_2$ concentration plumes LIF and PF compared well, but LIF was the only instrument capable of $SO_2$ measurements in the remote marine boundary layer due to its campaign limit of detection (LoD, 3 σ) of 0.07 ppb at 10 seconds compared with 0.4 ppb for the PF. Quantification of $SO_2$ using I-CIMS was challenging due to a significant interference, but good signal correlation with the other instruments was observed in polluted air masses. A comparison of response time was also made, for which the I⁻CIMS and LIF proved much faster than the PF with 3-efolding times of 0.6, 2 and 17 seconds respectively. This work demonstrates the importance of sensitive instrumentation like the LIF for quantifying low concentrations of $SO_2$, such as over remote marine environments, at the time resolutions required for a fast moving platform. This is particularly relevant now as a result of more stringent sulfur emission regulations for shipping, and likely more so in the future as anthropogenic $SO_2$ concentrations continue to decline.

## 1 Introduction

Sulfur dioxide ($SO_2$) plays a pivotal role in the chemistry of the troposphere, and has been long recognised as an anthropogenic air pollutant (Firket, 1936) and contributor to acid rain (Gorham, 1958), leading to a number of legislations limiting its emission. Since the 1970's, global anthropogenic $SO_2$ emissions have been decreasing (Smith et al., 2011) and are now below many countries' emission commitment limits (Department for Environment, 2024). However, even at present day levels, $SO_2$ from both anthropogenic and biogenic sources generate sulfate aerosols which still play an important role in the Earth's radiative budget (Capaldo et al., 1999; Myhre et al., 2013). In the atmosphere, $SO_2$ is oxidised by gas- and aqueous-phase





chemistry to sulfate aerosols, which contribute to the formation of cloud condensation nuclei (Faloona, 2009; Merikanto et al., 2009). Both the direct radiative forcing from these aerosols, and the indirect forcing from aerosol-cloud interactions result in a net cooling effect on the planet (Penner et al., 2001). Understanding the extent to which aerosol-cloud interactions are

masking greenhouse gas-induced warming remains the largest source of uncertainty in quantifying present day anthropogenic radiative forcing (Forster et al., 2021). In situ measurements of aerosols and their precursor species are necessary to reduce this uncertainty and to validate climate model estimations of radiative forcing. Therefore, it is of interest to accurately and precisely quantify the concentration of $SO_2$ in the background atmosphere if we are to predict the effects of changing emission rates on the climate.


The role of $SO_2$ in the formation of new particles is particularly important in clean environments with few primary particle sources, and $SO_2$ emissions from the global shipping sector represent an important anthropogenic source in remote regions. These have been declining over recent years as a result of regulations introduced by the International Maritime Organisation (IMO), limiting the sulfur content of ship's fuel. These measures were implemented in response to air quality concerns in

coastal regions, where aerosols from ship emissions are estimated to cause 400,000 premature deaths and ~ 14 million childhood asthma cases annually (Sofiev et al., 2018). The most recent regulation in January 2020, hereafter referred to as IMO2020, enforced a reduction of sulfur fuel content from 3.5 to 0.5 % by mass at the point of exhaust emission for ships in international waters. The global climate consequence of this regulation has been assessed by a recent surge of radiative forcing estimates ranging from 0.02 to 0.2 W m$^{-2}$ (Bilsback et al., 2020; Diamond, 2023; Gettelman et al., 2024; Jin et al., 2018; Jordan

and Henry, 2024; Quaglia and Visioni, 2024; Skeie et al., 2024; Sofiev et al., 2018; Yoshioka et al., 2024; Yuan et al., 2024, 2022), made using a number of different modelling methods and assumptions. Yoshioka et al., (2024) predicted a corresponding global mean warming of 0.04 °C averaged over 2020-2049, making it more difficult to limit warming to 1.5 °C, in line with the Paris Agreement, over the next few decades. Nevertheless, these regulations have resulted in significant reductions in atmospheric $SO_2$ concentrations over the ocean, and thus increased the relative importance of biogenic precursor

emissions such as dimethyl sulfide (Yang et al., 2016).

Measurements and models show that $SO_2$ in remote marine environments is usually on the order of 10 pptv (Bian et al., 2024). In order to test our understanding of $SO_2$ production and loss in these remote marine environments and over the range of altitudes where sulfate aerosol production is important, airborne sampling is required. Unfortunately, typical commercial

instruments currently used for the detection of $SO_2$ lack the sensitivity to perform these measurements at the time resolutions required for a fast moving platform. Aircraft studies using the pulsed fluorescence (PF) technique to measure $SO_2$ over the ocean are dominated by measurements of high $SO_2$ concentrations in ship plumes, mainly for assessing compliance to IMO regulations (Beecken et al., 2014; Lack et al., 2011; Yu et al., 2020). The only recent PF aircraft measurements of remote marine $SO_2$ were conducted by Zanatta et al., (2020), who struggled to quantify the low $SO_2$ concentrations seen at high

altitudes. Other remote marine $SO_2$ measurements via PF were performed at a stationary site, hence making use of long-term



averaging to achieve a limit of detection (LoD) of 25 pptv at 5 minutes in order to quantify the background levels as low as 50 pptv (Yang et al., 2016). However, these PF studies were both conducted pre-IMO2020 regulation. Alternative aircraft techniques used to measure $SO_2$ include the remote sensing technique of differential optical absorption spectroscopy (DOAS), which has again been reported for measurements of ship plumes (Berg et al., 2012; Cheng et al., 2019; Seyler et al., 2017).

The most recent measurements (post-IMO2020) using this technique were made by Mahajan et al., (2024) during a stationary site campaign to measure ship plumes, however, it was noted that $SO_2$ concentrations were below their LoD on a particular day due to sampling of clean air masses. Therefore, more specialised instruments with greater sensitivities are required for measurements of further declining $SO_2$ concentrations.

Chemical ionisation mass spectrometry (CIMS) measurements of $SO_2$ have been conducted on airborne platforms using a range of negative ion chemistries (Lee et al., 2018), with the best reported sensitivity coming from the use of a $CO_3^-$ ion by Thornton et al. (2002), Speidel et al. (2007) and Fiedler et al. (2009), achieving $3\sigma$ LoDs of ~ 1 pptv, 22 pptv and 30 pptv respectively at 1 s. More recently, an instrument that uses the technique of laser-induced fluorescence (LIF) to measure $SO_2$ has been developed by Rollins et al. (2016) and its performance on an aircraft has since been demonstrated on multiple field

campaigns (Rickly et al., 2021, 2022; Rollins et al., 2016, 2017). This LIF instrument has been reported to attain a LoD ($3\sigma$) of ~ 10.2 pptv at 1 s (Rickly et al., 2021) and can achieve a true 5 Hz measurement rate (Rollins et al., 2016). With comparably low LoDs to CIMS, LIF may be more favourable, especially for aircraft measurements of $SO_2$ due to its smaller size and weight, ease of operation, and lack of known interferences (Rickly et al., 2021). In this work, we introduce the University of York's custom-built LIF instrument, based on Rollins et al. (2016), for in situ trace measurements of $SO_2$, and compare airborne

measurements with both an iodide CIMS ($I^-$CIMS) and a commercial PF $SO_2$ analyser.

## 2 $SO_2$ instrumentation

The third Atmospheric Composition and Radiative forcing change due to the International Ship Emissions regulations (ACRUISE-3) campaign took place on 29th April to 3rd May 2022 aboard the UK Facility for Airborne Atmospheric

Measurements (FAAM) BAe-146 research aircraft (Yu et al., 2020). The campaign consisted of three 5-hour flights, spanning a range of altitudes between 0.07 and 3.2 km. The instrumentation available for measuring $SO_2$ during this campaign included the York LIF instrument, a PF $SO_2$ analyser, and an $I^-$CIMS which are described herein. Both individual ship plumes on the order of one to tens of ppb of $SO_2$ (termed 'polluted') and more remote marine regions outside of shipping lanes on the order of tens to a few hundred ppt (termed 'clean') were sampled in international waters around Milford Haven, UK and the Bay of

Biscay. Hence, these flights were ideal for comparing the three techniques as a wide range of concentrations were measured. In this work, the LoDs of the instruments are described to a $3\sigma$ confidence interval at 1 and 10 second averaging times and response time is defined as the time taken for 5 % of the initial concentration to remain, referred as the 3 e-folding response



time ($e^{-3}$). A summary comparing these statistical characterisations of the techniques as run during the ACRUISE-3 flights can be found in Table 1.


**Table 1.** Comparison of LIF, PF and I⁻CIMS techniques in terms of LoD, response time, sampling rate, and mixing ratio uncertainty as performed during the ACRUISE-3 campaign.

| Technique | Organisation | LoD at 10 (1) seconds (3 σ, ppbv) | 3 e-folding response time (seconds) | Sampling rate (Hz) | Uncertainty in mixing ratios (2 σ) |
|---|---|---|---|---|---|
| LIF | University of York | 0.07 (0.22) | 2 | 5 | 10 % + 6.5 pptv |
| PF | FAAM | 0.4 (1.1) | 17 | 1 | 18 % |
| I⁻CIMS | University of Manchester | - | 0.6 | 4 | - |

## 2.1 Laser-induced fluorescence

The University of York's laser-induced fluorescence (LIF) instrument is a custom-built system for the highly sensitive detection of $SO_2$, based on the system originally demonstrated by Rollins et al. (2016) and improved by Rickly et al. (2021). The fifth harmonic (216.9 nm) of an in-house built pulsed tuneable fibre-amplified semiconductor diode laser system (1084.5 nm, 3 ns pulse duration, 200 kHz repetition rate) is used to selectively excite $SO_2$, and the subsequent fluorescence photons are detected using a photon counting head (Hamamatsu H10682-210). The laser wavelength is tuned on and off a strong $SO_2$

transition ($\tilde{C}(^1B_2) \leftarrow \tilde{X}(^1A_1)$) peak, which is tracked using a reference cell that is maintained at a constant $SO_2$ concentration. The difference between the number of fluorescence photons at these positions is directly proportional to the $SO_2$ concentration within the sample cell.

The University of York laser system differs from that described in Rickly et al. (2021) in that we use a semiconductor optical
amplifier (SOA, Innolume) to pulse the continuous wave output of a distributed feedback seed laser diode (Innolume) at 200 kHz. The SOA is temperature controlled to 25 °C to ensure reproducible laser pulse generation. Other notable differences to Rollins et al. (2016) are that we use a proportional valve (Bürkert 2873) to maintain constant mass flow, and a pressure controller (Alicat PCH-100TORRA-D-MODTCPIP-A515) to maintain cell pressure.





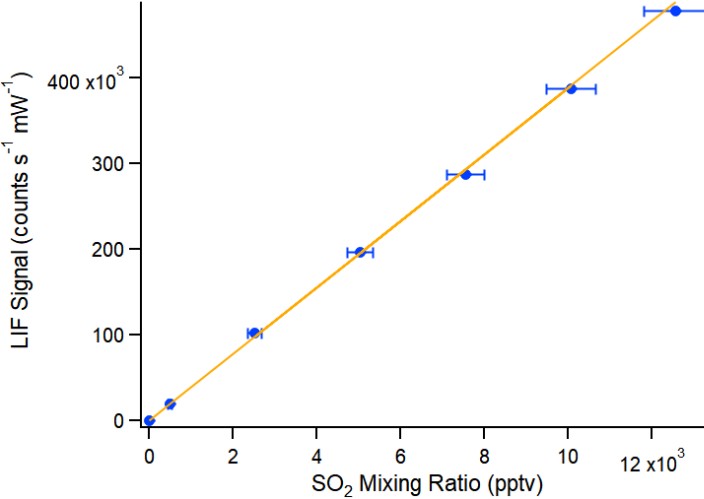


**Figure 1.** Example laboratory $SO_2$ calibration in zero air where the LIF signal is the difference between the linearised and normalised on-line and off-line sample cell counts. The orange line shows a York regression fit to the seven data points, indicating a slope of $38.7 \pm 2.1$ counts s$^{-1}$ mW$^{-1}$ pptv$^{-1}$, a y-intercept of $620 \pm 1700$ counts s$^{-1}$ mW$^{-1}$ (both 2 σ confidence), and a correlation coefficient of $R^2 = 1.0$. The x error bars are dominated by the uncertainty in the $SO_2$ calibration standard ($\pm 5$ %), but also include uncertainties in the mass flow controllers
($\pm$(0.8 % of reading + 0.2 % of full scale)) and the cell flow meter ($\pm 3$ %). The y error bars (not visible) represent two standard errors of the LIF signal.

The pulse pair resolution of the photon counting head detector (20 ns) limits the available signal count rate to be equal to the repetition rate of the excitation laser, resulting in a need for a linearity correction that is important at high signal rates (Rollins
et al., 2016). However, due to the combined effect of relatively low laser power (see below) and high laser repetition rate (200 kHz) in this work, the correction was only effective mainly for ship plume measurements where mixing ratios reached 140 ppb. Linearised counts are then normalised by laser power, which is measured by a phototube (Hamamatsu R6800U-01). The difference between the corrected fluorescence counts at the on-line and off-line laser wavelength positions is converted to $SO_2$ mixing ratio via the sensitivity factor of the system (Equation 1), which is determined as the slope of a calibration plot using a
dynamic dilution of an $SO_2$ standard in air. An example of a multi-point calibration from a laboratory experiment is shown in Fig. 1.

$$[SO_2] = \frac{S_{online} - S_{offline}}{C_{SO_2}} \tag{1}$$

where $[SO_2]$ is the $SO_2$ mixing ratio (pptv), $S_{online}$ and $S_{offline}$ are the on-line and off-line linearised and normalised fluorescence counts (counts s$^{-1}$ mW$^{-1}$), and $C_{SO_2}$ is the experimentally determined sensitivity factor of the system to $SO_2$ (counts
s$^{-1}$ mW$^{-1}$ pptv$^{-1}$).





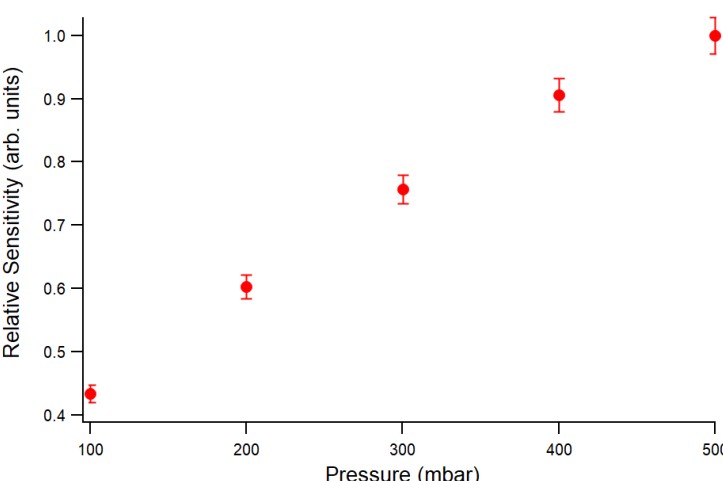

**Figure 2.** Relative LIF sensitivity factor over a cell pressure range of 100 to 500 mbar. The error bars are derived from the sensitivity uncertainties (Fig. 1), given to two standard errors.

Figure 2 shows how the relative sensitivity (calculated via normalising the sensitivity factor at a given pressure by that at 500 mbar) varies with cell pressure for the York LIF system. Whilst the number of $SO_2$ molecules per unit volume increases by a factor of 5 from 100 to 500 mbar, the sensitivity factor only increases by approximately a factor of 2 due to the increasing importance of quenching at higher pressures. During the ACRUISE-3 flights, the use of a pressure-building ram inlet allowed both the sample and reference cells to be operated at $400 \pm 2$ mbar for the full altitude range (between 0.07 and 3.2 km) of the campaign to maximise instrument sensitivity (Fig. 2) at the lower external pressures.





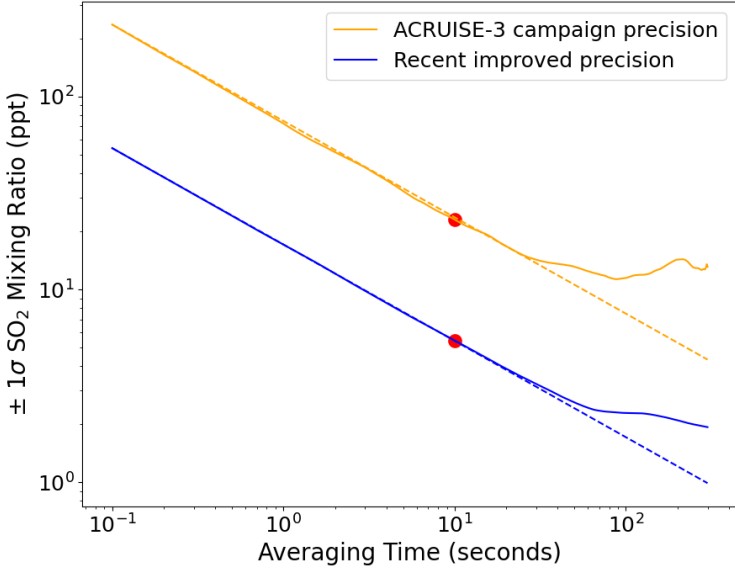

**Figure 3.** LIF precision (Allan deviation) of a 10 min stable ambient measurement of mean mixing ratio 75 pptv during flight C287 (orange) and a 3.5 h zero air measurement, performed recently in the laboratory with improved laser power and sensitivity (blue). Both traces are compared to the expected precision (Poisson limit). The precision (1 σ) at an averaging time of 10 s for each trace is marked by the red dots (23 pptv for the orange trace, 5.4 pptv for the blue).

The ACRUISE-3 aircraft campaign was the first deployment of the York LIF instrument in the field. At the time, we were facing difficulties with lower fifth harmonic laser power (mean power of ~ 27 μW during the campaign) compared to Rollins et al. (2016) (~ 1 mW), using a comparable optical setup. Therefore, the 3 σ LoD was determined as 70 pptv at 10 seconds from a stable ambient measurement of mixing ratio close to the LoD, which is shown by the orange Allan deviation trace in Fig. 3.

**Table 2.** Summary of the calibrations performed in both ambient air and zero air during each flight, showing the mean sensitivity ($\bar{x}$), standard deviation of the sensitivities ($\sigma$) and the number of calibrations ($N$).

| Flight | Ambient Air | | | Zero Air | | |
|---|---|---|---|---|---|---|
| | $\bar{x}$ (counts s$^{-1}$ mW$^{-1}$ pptv$^{-1}$) | $\sigma$ (counts s$^{-1}$ mW$^{-1}$ pptv$^{-1}$) | $N$ | $\bar{x}$ (counts s$^{-1}$ mW$^{-1}$ pptv$^{-1}$) | $\sigma$ (counts s$^{-1}$ mW$^{-1}$ pptv$^{-1}$) | $N$ |
| C285 | 33.8 | 0.7 | 9 | 34.0 | 1.7 | 10 |
| C286 | 32.7 | 1.5 | 7 | 34.8 | 0.9 | 7 |
| C287 | 34.8 | 0.8 | 14 | 35.7 | 1.3 | 4 |



During the ACRUISE-3 flights, multi-point calibrations were carried out using a 5 ppm $SO_2$ in $N_2$ standard (BOC, ± 5 %) added to the end of the inlet across the expected concentration range (0.5 – 12.5 ppb) approximately every 30 minutes to ensure data accuracy and to capture instrumental drift. A flow rate of 2 SLPM was used, giving a 3 e-folding response time of ~ 2 seconds. To assess the possible quenching effect of excited $SO_2$ by water vapour, or increased wall losses when sampling humid air, calibrations in both stable ambient air and dry zero air were carried out, for which these effects proved negligible as shown in Table 2. For calibrations in zero air, it was necessary to overflow the inlet, however, subsequent analysis deemed this overflow insufficient for a true zero to be measured, likely a result of pressure build-up in the inlet line from the ram inlet. Since these calibrations were performed during stable ambient mixing ratios in absence of any spikes in $SO_2$, such as the conditions shown in Fig. 12, the calculated sensitivities are unlikely to be affected. Hence, we justify including the calibrations in zero air in our analyses. As a result of inconsistencies in the laser linewidth, the sensitivities were seen to vary slightly during the course of a flight (Fig. S1 – S3), and hence a mean sensitivity has been calculated from both calibrations in ambient and zero air, and applied on a flight-by-flight basis. Finally, the uncertainty in the $SO_2$ mixing ratios was calculated from the uncertainty in the instrument sensitivity via a York regression fit to a calibration plot (Wu and Zhen Yu, 2018, Fig. 1). This gives a 2 σ uncertainty of ~ 10 % + 6.5 pptv across each flight.

Since the ACRUISE-3 campaign, improvements have been made to the York LIF system. Firstly, the conversion efficiency of the fifth harmonic generation has been improved by replacing the KTP crystal with a temperature-controlled PPLN crystal, yielding a ~ 30-fold increase in UV laser power. In addition, the efficiency of the collection optics has also been improved by adding a lens (Edmund, #49-695) before the PMT module to focus the fluorescence into the detection head, giving a factor of ~ 4 better sensitivity. These advancements have substantially improved the instrument's precision, as shown by the blue Allan variance trace in Fig. 3.

## 2.2 Pulsed fluorescence

As part of the core instrumentation on board the FAAM aircraft, a commercial Thermo Fisher Scientific model TEi-43i TLE $SO_2$ analyser was used to measure $SO_2$ during the entire ACRUISE campaign. Based on the UV pulsed fluorescence (PF) method, it uses a broader, non-tuneable $SO_2$ excitation wavelength range compared to LIF and hence is more susceptible to interfering species. To minimise $SO_2$ fluorescence quenching by water vapour, the PF is equipped with an external Nafion dryer (PermaPure Multi-strand, PD-50T-24MPR). A heated hydrocarbon kicker is used to remove interferences caused by volatile organic compounds which fluoresce at similar UV wavelengths to that of $SO_2$. Owing to their broad spectral features relative to $SO_2$, these compounds do not interfere with $SO_2$ mixing ratio measurements using the LIF technique, as they contribute equally to background counts at both the on-line and off-line wavelengths.





Modifications were made to the PF instrument in 2016 to improve its suitability for airborne measurements. The sample flow
rate was increased from approximately 0.5 to 2 SLPM to improve the instrument response time by replacing the original TEi43i
glass capillary and flow sensor with a mass flow controller (MFC3, Alicat Scientific, MCS-5SLPM-D-I-VITON). Also, a
second hydrocarbon kicker was added to enhance sample flow conductance at this higher sample mass flow rate.

During the flights, the PF was run at a flow rate of 2 SLPM, giving an in-flight response time (3 e-folding) of 17 seconds. A
210 comparable inlet to the LIF instrument was used. In-flight single point calibrations were carried out by overflowing the
instrument inlet with a 2.5 SLPM mass flow controlled calibration gas mixture of 374 ppb $SO_2$ in Air (BOC, ± 6 %). Mulit-
point calibrations were also performed on the ground post-ACRUISE-3 deployment as a check of the sensitivity. To account
for baseline drift, frequent (~10 to 15-minute interval) zero measurements were performed by passing the air sample through
an external zero air scrubber cartridge filled with activated charcoal. Mean zeros are then linearly interpolated to provide a
215 drift-corrected baseline, which is subtracted from the raw fluorescence counts, before being scaled by the detector sensitivity.

The instrument LoD (3 σ) during the ACRUISE-3 deployment has been determined as 400 ppt at 10 seconds. Finally, the
overall uncertainty in $SO_2$ mixing ratios has been calculated as ± 18 % for a 2 σ confidence interval.

**2.3 Iodide chemical ionisation mass spectrometry**

Matthews et al., 2023 have previously described in detail the University of Manchester (UoM) iodide ion- High Resolution-
Time of Flight-Chemical Ionisation Mass Spectrometer (I⁻CIMS, Aerodyne Research, Inc) for use on the FAAM Research
Aircraft. Briefly, iodide ions cluster with sample gasses in the pressure-controlled ion-molecule reaction (IMR) region creating
a stable adduct. The flow is then sampled through a critical orifice into the first of the four differentially pumped chambers in
the CIMS, the short segmented quadrupole (SSQ), which is also independently pressure controlled. Quadrupole ion guides
transmit the ions through these stages. The ions are then subsequently pulsed into the drift region of the CIMS where the arrival
time is detected with a pair of microchannel plate detectors with an average mass resolution of 4000 (m/Δm). The UoM I⁻
CIMS operates with an IMR pressure of 72 mbar for aircraft campaigns and instrument backgrounds are taken every minute
for 6 seconds by overflowing the inlet with ultra-high purity (UHP) nitrogen. The CIMS instrument analysis software (ARI
Tofware version 3.1.0, Stark et al., 2015) was utilized to obtain high resolution, 1 Hz, time series of the compounds presented
here. Mass-to-charge calibration was performed for 5 known masses; I-, I-.H2O, I-.HCOOH, I2-, I3-, covering a mass range of
127 to 381 m/z. The mass-to-charge calibration was fitted using a square-root equation and was accurate to within an average
of 1 ppm.



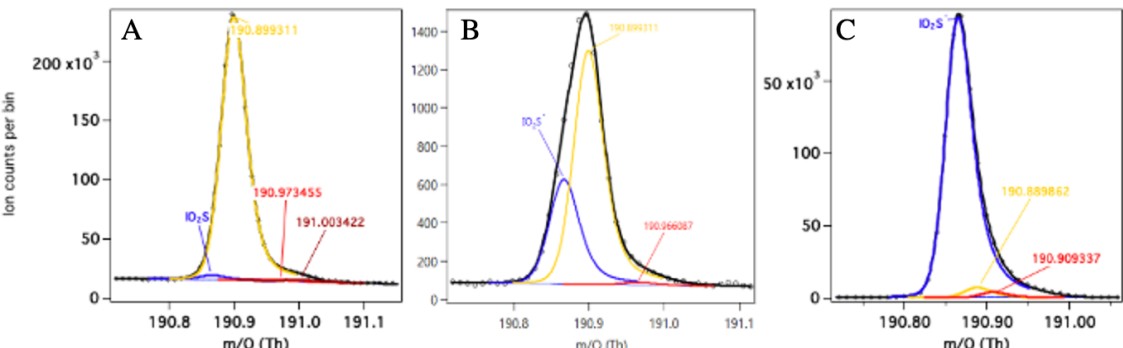

**Figure 4.** High resolution peak fitting at m/z 191 during ambient sampling in ACRUISE-3 for A) average mass spectrum for flight C285, B) in a ship plume and C) calibration using a commercial standard.

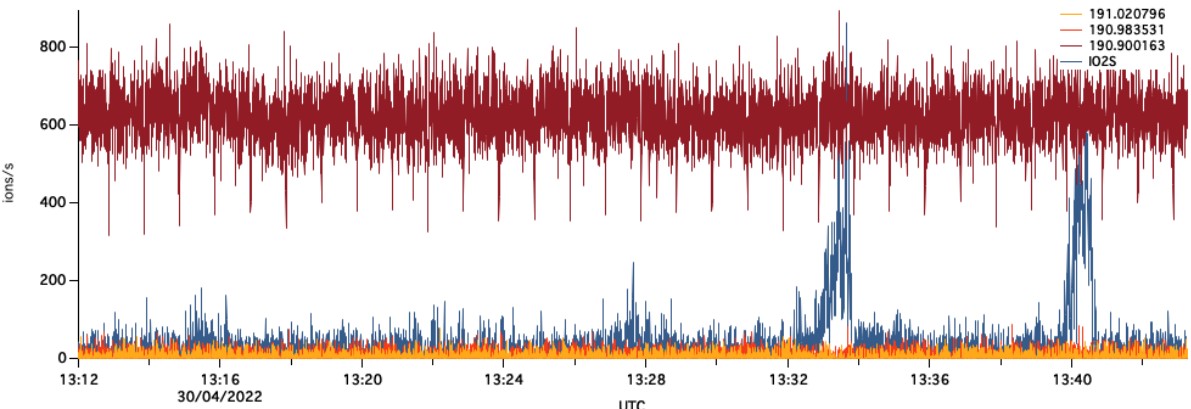

**Figure 5.** Time series of the high-resolution peak fitting at m/z 191 taken at 4 Hz. Peaks at ~ 13:33 and 13:40 are due to ship plumes.

I-CIMS detects $SO_2$ as a cluster with iodide at m/z 190.866372. During the ACRUISE-3 campaign, the $SO_2$ peak is in close proximity to a larger interfering peak, as shown in Fig. 4. The UoM I⁻CIMS has sufficient resolving power to accurately separate the overlapping peaks. Diagnostic tools in the analysis software (ARI Tofware version 3.1.0, Stark et al., 2015) have been used to estimate the uncertainties in the signal intensity fitted for $SO_2$ due to the mass calibration and multipeak fitting. However, despite the very accurate mass calibrations there is still an associated uncertainty of approximately 30 % for the signal intensity from an offset of 1 ppm. Additional uncertainties arise from the multipeak fitting at m/z 191 and for $SO_2$ is an average of 97 % when measuring in ambient air (Fig. 5). This uncertainty however decreases in ship plumes as the magnitude of the $SO_2$ peak increases. In comparison, the uncertainty in the signal fitted for $SO_2$ during the offline calibration is significantly reduced and is 3 % for the multipeak fitting.



The I⁻CIMS SO₂ measurements were calibrated offline using a stable flow of SO₂ generated using a commercially sourced
known concentration gas mixture (BOC, 1 ppm ± 5 % of SO₂ diluted in air) and a custom-built dynamic dilution system, which
allows for a calibration gas to be diluted into a carrier gas, and in this instance, ultra-high purity (UHP) $N_2$ was used. The
concentration of the outflow can be controlled by varying the flows of the calibrant and carrier gas, each of which are
individually regulated using two MFCs (Alicat Scientific, MCS-5SLPM-D-I-VITON and MCS-500SCCM-D-I-VITON). In
this case 1.2 SLM of the outflow was delivered to the CIMS by 1⁄4" PTFE tubing and the overflow exhausted at calculated
concentrations of 25, 50, 75, and 100 ppb. Additionally, the instrument's humidity dependence to the detection of SO₂ was
calculated by actively adding water vapour into the IMR by passing UHP $N_2$ through deionised water (Fig. 6). The presence
of I•H₂O- clusters, which are proportional to humidity of the air as it enters the instrument, can alter the ionisation efficiency
of species detected by I⁻CIMS and for SO₂ results in decreased sensitivity with increasing I.H₂O- clusters (i.e. negative
humidity dependence). Inconsistencies in the calibration results with measured concentrations in the field revealed an
unquantifiable offset in the retrieved SO₂ data, for this reason the CIMS data will be presented in cps and utilised to assess the
varying time response of the three instruments.

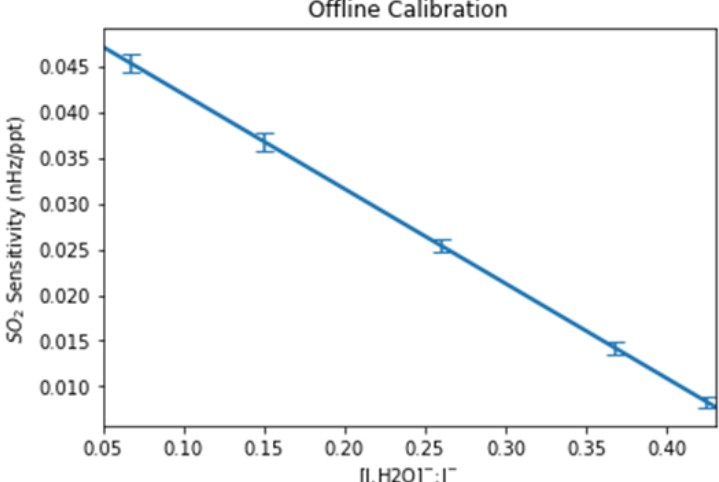

**Figure 6.** UoM I⁻CIMS humidity dependent sensitivity to SO₂ determined from an offline calibration using a commercial standard of SO₂.
The error bars represent one standard deviation.

## 3 Results and discussion

This work compares airborne SO₂ measurements from LIF, PF and I⁻CIMS based on their measured time series, instrument
precision and response time during the ACRUISE-3 campaign. To construct a consistent dataset from all three instruments,





each time series has been averaged to 10 seconds (by taking the mean of each 10 s block). This averaging time was chosen as

a balance between the rapid changes observed due to the direct sampling of ship plumes and the response time of the PF system

of 17 seconds. The comparison is split between polluted (high $SO_2$) and clean (low $SO_2$) regions due to the different analytical

requirements for measurements in these distinct environments. A map showing all three flight tracks, coloured by the LIF $SO_2$

mixing ratios, is shown in Fig. S4.

**3.1 Polluted environments**

Measurements made in polluted $SO_2$ marine environments include those in individual ship plumes and within shipping lanes.

An example of a polluted time series comparison from flight C285 is shown in Fig. 7 which contains a series of time-matched

ship plume events. Comparing the magnitude of these peaks suggests that the LIF and PF agree well. These conclusions are

consistent with the correlation plots containing data from all three ACRUISE-3 flights. Figure 8A shows the correlation

between the LIF and PF from 400 pptv (the LoD of the PF at 10 s) to the greatest plume mixing ratio of ~ 40 ppb, and shows

near-unity agreement (slope = 0.90) to within the combined uncertainty of the LIF and PF. The relationship between the CIMS

data in cps and LIF $SO_2$ mixing ratios also displays a linear correlation (Fig. 8B) but with a poorer fit. Due to the challenges

in calibrating the CIMS, described above, the gradient of this plot has been used to estimate CIMS $SO_2$ mixing ratios for

comparison of response time and noise distributions across the three $SO_2$ measurement techniques.

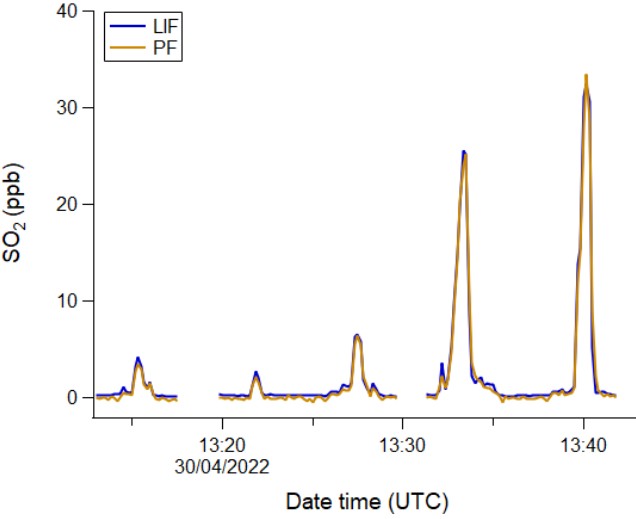

**Figure 7.** Time series of 10 s averaged, time matched data during flight C285, comparing LIF and PF $SO_2$ measurements.




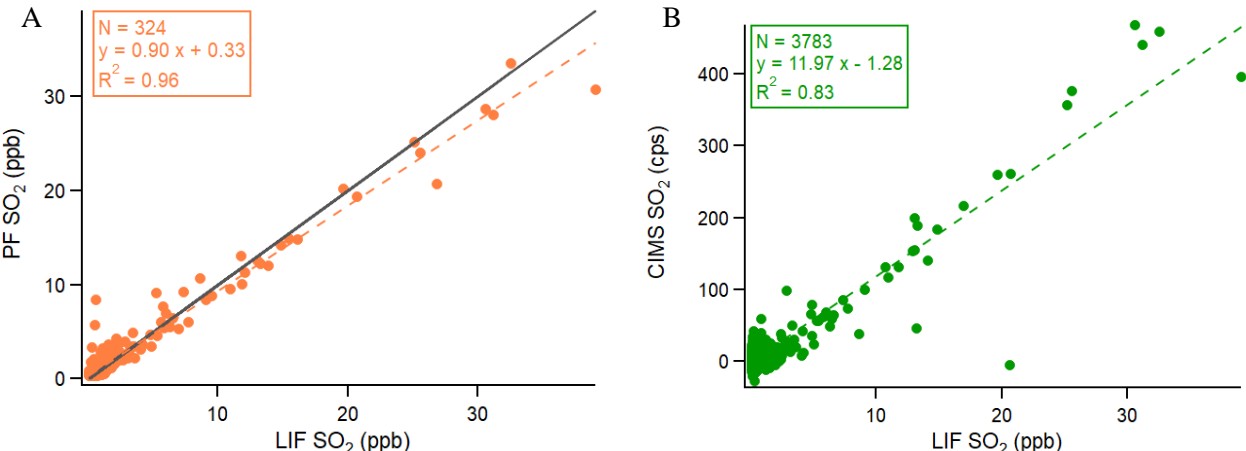

**Figure 8.** Correlation of the 10 s averaged, time matched data for all three ACRUISE-3 flights. (A) Correlation between the PF and LIF SO$_2$ mixing ratios, excluding points below the instrument's LoD. (B) Correlation between the I$^-$CIMS in cps and LIF SO$_2$ mixing ratios. The coloured dashed lines represent the linear fit and the solid black line represents the 1:1 ratio (for the mixing ratio comparison only).

A comparison of response time can be made from the time series of a single ship plume peak in Fig. 9, recorded at 5 Hz, 1 Hz and 4 Hz for the LIF, PF and I$^-$CIMS respectively. To remove the lag time as a result of different inlet lengths, the peaks have been time-matched by the increase in SO$_2$ mixing ratios upon intersecting a ship plume. Figure 9 shows the SO$_2$ mixing ratios recorded by the I$^-$CIMS fall to out-of-plume levels the quickest and its time series displays the greatest structure. This is evident in the faster I$^-$CIMS response time of 0.6 s (3 e-folding) compared to the LIF of 2 seconds and the PF of 17 seconds. The LIF signal shows similar structure to the I$^-$CIMS data, but the slower gas flush rate means the features are smoothed and the LIF takes longer than the I$^-$CIMS to return to background levels. Further improvements to the LIF system, detailed in section 2.1, have increased the LIF true measurement rate to 10 Hz. The comparatively slow response time of the PF instrument results in it being unable to resolve the plume structure, and shows a significant delay in returning to background levels.



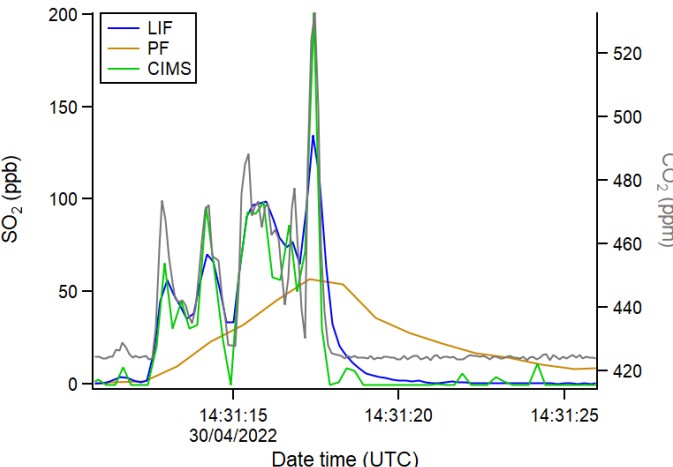

**Figure 9.** Time series comparing instrument response time, matched by the increasing $SO_2$ mixing ratio due to measurement of a ship plume during flight C285. The LIF data is presented at 5 Hz, the PF at 1 Hz, the I⁻CIMS at 4 Hz and the $CO_2$ data at 10 Hz. To note, I⁻CIMS $SO_2$ mixing ratios have been estimated based on the gradient of the fit to the LIF data (Fig. 8B).

Plume sampling can be used to quantify emissions ratios, and the ratio of $SO_2$:$CO_2$ can be used to calculate ship sulfur fuel content for applications in compliance monitoring (Beecken et al., 2014; Berg et al., 2012; Kattner et al., 2015; Lack et al., 2011; Yu et al., 2020). Average emissions ratios can be derived via two methods for a ship plume event: a) the integration method where the area under the peak and above the baseline is calculated (Beecken et al., 2014; Kattner et al., 2015; Lack et al., 2011; Yang et al., 2016), and b) the regression method where the slope of a correlation plot is taken (Aliabadi et al., 2016). The regression method is a useful approach as it removes the necessity to subtract the background concentrations (Wilde et al., 2024). However, it is a less popular approach, especially for in situ airborne sampling, since it relies on sufficient data points for a statistically valid regression analysis and a similar plume structure between the two species, as shown by the $SO_2$ I⁻CIMS and $CO_2$ trace in Fig. 9. Hence, this requires fast response time instrumentation for aircraft measurements of plume transects as the plume duration can be extremely short. On the other hand, the integration method provides a response time-independent way of reliably analysing these short duration plume events.





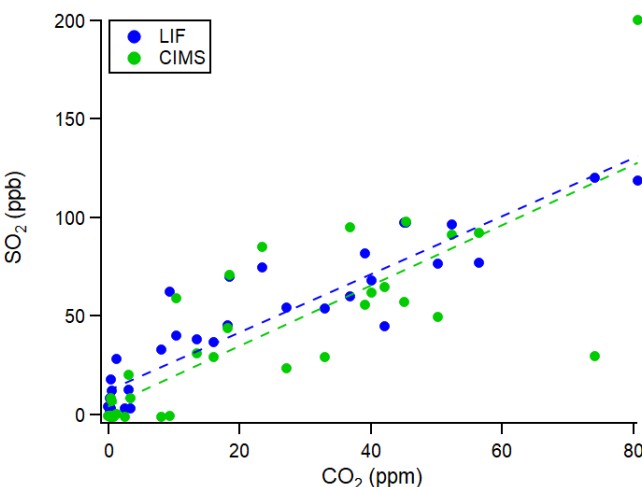

**Figure 10.** Regression analysis to calculate the $SO_2:CO_2$ emission ratio (gradient) for the ship plume during flight C285. The linear fits have gradients and $R^2$ values of $1.48 \pm 0.21$ ppb ppm$^{-1}$ and 0.85 (LIF); 1.53 ppb ppm$^{-1}$ and 0.65 (CIMS).

This work makes use of the fast response time of the I$^-$CIMS (to achieve fine plume structure) and the accuracy of the LIF to compare the integration and regression methods. Therefore, I$^-$CIMS $SO_2$ mixing ratios have been estimated based on the gradient of the fit to the LIF data (Fig. 8B). In the integration method, background concentrations have been determined using an exponentially weighted moving average, and peak areas have been calculated relative to this baseline using a trapezoidal approximation. For the plume in Fig. 9, the corresponding regression analysis is shown in Fig. 10 and the results of this ship

plume event, as well as three others, are shown in Table 3. Plots for the remaining three plumes are given in Fig. S5 – S7. To note, due to the much slower response time of the PF technique, an emission ratio via the regression method could not be calculated.







**Table 3.** Comparison of integration and regression methods for calculating $SO_2$:$CO_2$ emission ratios of four ship plumes during different flights. The LIF and I-CIMS $SO_2$ data and the $CO_2$ data have been sampled to 4 Hz, however, the PF $SO_2$ data is at 1 Hz. *I-CIMS $SO_2$ mixing ratios have been estimated based on the gradient of the fit to the LIF data (Fig. 8B). Since no instrument uncertainty was reported, an estimation of errors has not been made for these emission ratios. All other emission ratio uncertainties have been given to a 2σ confidence interval. The corresponding sulfur fuel content (SFC) has been calculated via Equation 2 and is given in brackets after each emission ratio.

| Flight | Plume duration (seconds) | Number of data points | $SO_2$:$CO_2$ emission ratio (ppb ppm$^{-1}$) via integration (SFC, %) | | | $SO_2$:$CO_2$ emission ratio (ppb ppm$^{-1}$) via regression (SFC, %) | |
|---|---|---|---|---|---|---|---|
| | | | LIF | PF | I-CIMS* | I-CIMS* | LIF |
| C285 | 9 | 36 | 2.08 ± 0.05 (0.48 ± 0.01) | 2.19 ± 0.11 (0.51 ± 0.03) | 1.73 (0.40) | 1.53 (0.36) | 1.48 ± 0.21 (0.34 ± 0.05) |
| C286 | 7 | 28 | 1.24 ± 0.06 (0.29 ± 0.01) | | 1.91 (0.44) | 1.85 (0.43) | 0.74 ± 0.13 (0.17 ± 0.03) |
| C287 | 59 | 236 | 1.95 ± 0.03 (0.45 ± 0.01) | 2.00 ± 0.06 (0.46 ± 0.01) | 1.97 (0.46) | 2.11 (0.49) | 1.38 ± 0.12 (0.32 ± 0.03) |
| C287 | 156 | 624 | 1.78 ± 0.03 (0.41 ± 0.01) | 1.70 ± 0.04 (0.39 ± 0.01) | 2.22 (0.52) | 2.29 (0.53) | 1.44 ± 0.07 (0.33 ± 0.02) |

A check of the compliance of the sampled ships to the IMO2020 regulation (0.5 % sulfur fuel content in international waters) can be made through calculation of the sulfur fuel content (SFC) from the emission factors. Assuming that 87 % of ship fuel by mass is carbon, the SFC mass percent is related to the emission ratio via the following equation (Kattner et al., 2015):

$$SFC[\%] = \frac{SO_2[ppb]}{CO_2[ppm]} \times 0.232[\%] \qquad (2)$$

where the $SO_2$:$CO_2$ ratio is the emission ratio, calculated above, and 0.232 is the mass conversion factor for fuel content.

Equation 2 also assumes that all sulfur is emitted as $SO_2$ and all carbon as $CO_2$. The latter of these assumptions is deemed a reasonable estimate since little to no CO was measured during the ACRUISE-3 campaign, suggesting a complete combustion pathway. However, it is known that not all sulfur is released as $SO_2$ – some is directly emitted as sulfate, $SO_4^{2-}$. The amount of $SO_4^{2-}$ released has been shown to correlate with the SFC (Yu et al., 2020) and it also increases with plume aging. Since the calculation of plume age is beyond the scope of this paper, the SFC is not corrected for sulfate (which is estimated as 6 % for

a plume age of max. 15 minutes (Yu et al., 2020)), and this 6 % discrepancy has been included in the uncertainty. Therefore, using the mean of the emission ratios calculated via the two methods, the corresponding SFCs of the sampled ships are given in Table 3, which shows they are all compliant to the IMO2020 regulation.

As shown in Table 3, the accuracy of the regression method for calculating emission ratios is dependent on the response time
of the instrument. The I-CIMS has a response time closer to that of the $CO_2$ measurements and hence gives emission ratios that are in greater agreement to those calculated via the integration method. The consistently lower regression emission ratios




derived from the LIF data are likely due to the smoothing effect as a result of a longer response time compared to the I⁻CIMS. However, the I⁻CIMS is limited by its large uncertainty and its greater $R^2$ values suggest that a single factor correction is unable to capture any variability associated with the interference. Therefore, improvements to the LIF's system response time will

make it the more desirable instrument for calculating emission ratios via the regression method.

### 3.2 Clean environments

In order to understand the role of sulfur in controlling particle formation in remote regions, and thus their direct and indirect climate impacts, measurements of $SO_2$ need to be made with sufficient sensitivity and precision to provide observations at a

time resolution capable of constraining on our understanding of key processes. A comparison of precision and noise distribution has been made for the three instruments from low $SO_2$ mixing ratios measured in clean marine environments, outside of shipping lanes. Figure 11A shows an example time series comparison of clean $SO_2$ measurements from flight C286. Two observations can be made from this time series comparison which are more evident in the corresponding histogram plot in Fig. 11B: (1) $SO_2$ is above the LIF LoD, unlike the PF and I⁻CIMS and (2) the distribution of the LIF is much narrower than

the PF and I⁻CIMS, given their standard deviations of 0.044, 0.14 and 0.27 ppbv respectively. For exploring statement (1), it is necessary to consider the 3 σ LoDs of the three instruments during the flight at the same averaging time as the time series and each other (10 s). These are 0.07 and 0.4 ppbv for the LIF and PF respectively at 10 s, and even higher for the I⁻CIMS. Comparing the mean mixing ratios measured by the LIF, PF and I⁻CIMS of 0.18, -0.094 and 0.050 ppbv, it is evident that the LIF is sensitive enough to capture these low mixing ratios. This is not the case for the PF and I⁻CIMS and therefore their

observed distributions are indistinguishable from a measurement of zero.

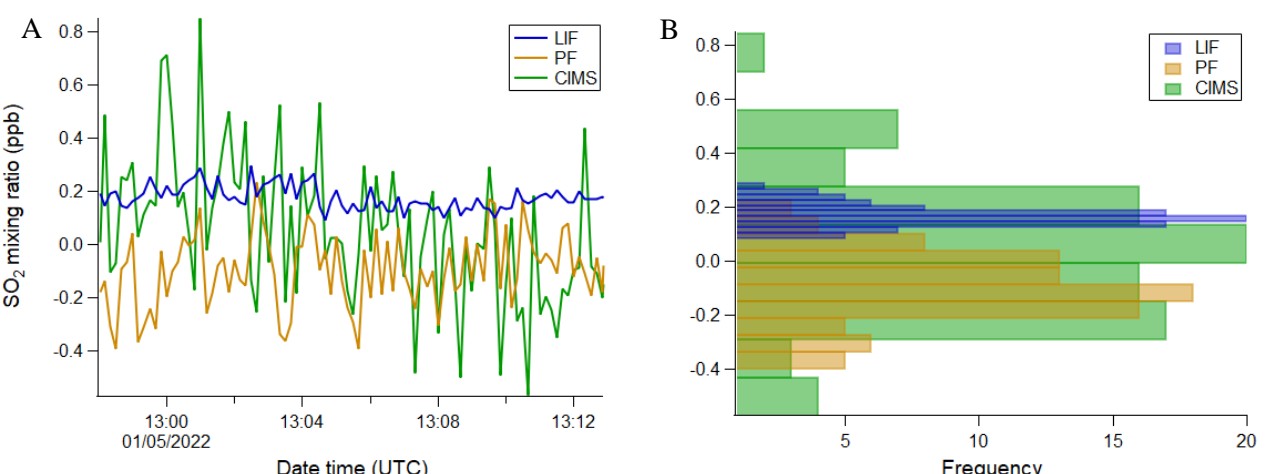

**Figure 11.** Time series (A) and histogram (B) of 10 s averaged, time matched data during flight C286, comparing the LIF, PF and I⁻CIMS $SO_2$ measurements. The distributions are divided into 10 bins for each instrument and hence the bin widths are scaled accordingly.

 


The 15-minute time period in Fig. 11A has been chosen as the ambient measurements are relatively stable and hence useful for comparing instrumental noise in flight via the width of the distributions in Fig. 11B. We have assumed a well-mixed atmosphere and that the LIF distribution is determined predominantly by instrumental noise (evidenced by Fig. S8). Therefore, as ambient variability is present but minimal, the LIF distribution is an upper limit assessment of its precision. We conclude

that the noise distribution for the LIF is significantly narrower than the PF and I$^-$CIMS, making it a powerful tool for capturing $SO_2$ mixing ratios in remote marine environments at a range of altitudes. Figure S9 shows an altitude plot of LIF $SO_2$ mixing ratios averaged across each 100 m bin width for all three ACRUISE-3 flights. This data is compared to $SO_2$ mixing ratios measured by the LIF during the seventh aircraft campaign of The North Atlantic Climate System Integrated Study (ACSIS-7), which sampled clean remote marine air over the North Atlantic during 5$^{th}$ – 9$^{th}$ May 2022.

**4 Conclusions**

Three $SO_2$ instruments were involved in an intercomparison experiment on board the U.K. FAAM research aircraft: LIF, PF and I$^-$CIMS. A range of $SO_2$ concentrations were measured, from < 70 ppt in clean marine environments up to 40 ppb in ship-polluted environments (at 10 seconds averaging), west of the English Channel over international waters. In polluted environments, $SO_2$ measurements made by the LIF and PF agreed within the errors of the instruments. However, the LIF and

I$^-$CIMS measurements were found to disagree by a constant factor. This work has also allowed a comparison of $SO_2$:$CO_2$ emission ratio calculation methods (integration versus regression), and the regression method has proved to be in greater agreement to the integration method when using fast time response instrumentation. In clean environments, the ambient mixing ratios are below the LoDs of the PF and I$^-$CIMS instruments, and therefore the LIF is the only instrument in this study able to detect $SO_2$ mixing ratios between 70 and 400 ppt. From this intercomparison, we conclude that for measurements of low $SO_2$

concentrations requiring high sensitivity and low noise, such as those in remote marine environments, LIF is a powerful technique. While I$^-$CIMS has demonstrated a response time approximately three times faster than LIF, making it more suitable for aircraft measurements, its sensitivity to $SO_2$ is limited and for the instrument used in this study, has unresolved issues surrounding accurate quantification. Ongoing improvements to the LIF have already improved the instrument LoD to 16.2 pptv at 10 s (3σ) and are increasing its flush rate towards flux-scale response times (>5 Hz), therefore allowing both fast and

accurate measurements of $SO_2$. All three techniques are valuable for improving our understanding of atmospheric $SO_2$, but application dependent. The LIF technique is becoming more crucial both today and in the future, as more stringent emission reductions, such as the IMO2020 regulation, lead to cleaner $SO_2$ environments.

*Code and data availability*

The LIF and I$^-$CIMS data in addition to the code to reproduce the figures in this paper can be found at https://github.com/wacl-york/LIF-cals (link will be made available on publication). Aircraft measurements including the PF data are available on the



CEDA Archive in the individual flight folders from the ACRUISE (https://catalogue.ceda.ac.uk/uuid/d6eb4e907c124482881d7d03c06903e4) and ACSIS (https://catalogue.ceda.ac.uk/uuid/7e92f3a40afc494f9aaf92525ebb4779) projects.


*Author contribution*

The York LIF instrument was based on the work of AR and built by SY, LT, JV, SR, EG and PE, and supervised by PE. Aircraft LIF measurements were taken by LT and JV, I⁻CIMS by EM and TB and PF by SB and SB. The ACRUISE aircraft campaign was led by MY with planning from JL, HC and DP. LT performed the data analysis with help from SL and DP. The

paper was written by LT (with the I⁻CIMS section written by EM), with contributions from all co-authors.

*Competing interests*

The authors declare that they have no conflict of interest.

*Acknowledgements*

The authors would like to thank the UK Natural Environment Research Council (NERC) for supporting this work via capital award NE/T008555 for the development of the York LIF-SO2 and supporting the fieldwork via the ACRUSIE project (NE/S005390) and the NERC National Centre for Atmospheric Science (NCAS) for supporting some of the remote ocean flights as part of the North Atlantic Climate System Integrated Study (ACSIS) program of work. LT would like to thank the

Panorama NERC Doctoral Training Partnership (NE/S007458). EM would like to thank Harald Stark for his advice on resolving and calibrating the CIMS SO$_2$ peak.

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
