# Peer review of "An intercomparison of aircraft sulfur dioxide measurements in clean and polluted marine environments"

_EGUsphere, 2025_

## Author Comment (AC1)

**Response to Referee 1**

We would like to thank the referee for their kind comments on the clarity and readability of the paper and also for their feedback. We note the referees comment regarding the need to motivate the use of LIF to detect $SO_2$ in the introduction and address that in our response to the comment about line 55.

Line 13: add a word after "(PF)" to indicate that this is not a form of mass spectrometry. Perhaps "instrument"?

We have added the word "analyser" after PF (unless referring to PF data, PF signal, PF technique etc.) and the word "instrument" after LIF, for clarity.

Line 17: ppb is used where as elsewhere "pptv" is used. Be consistent please.

We have amended every instance of "pptv" or "ppbv" to "ppt" and "ppb" respectively, in line with the standard convention in this field.

Line 55: I think at the end of this paragraph a short new paragraph would be helpful outlining the properties of $SO_2$ that make it suitable for detection – it's cross section, stability etc. Here it would be helpful to explain how UV remote sensing instruments are not suitable for estimating remote $SO_2$ abundances due to their low concentrations, for example, hence motivating the focus of this work. This comment relates to the overall statement: "I found that it lacked a bit in the introduction to motivate what it is about $SO_2$ that lends LIF as being such a useful approach to its detection".

We have added the following text in the specified location:

"$SO_2$ absorbs strongly in the ultraviolet region of the electromagnetic spectrum, characterised by a series of absorption bands at wavelengths from 403 to 106 nm (Manatt et al., 1993, Stark et al., 1999; Rufus et al., 2003). Therefore, spectroscopic $SO_2$ detection techniques typically exploit this UV region.

Fluorescence-based techniques employ excitation in the $\tilde{C}$ band (~190-230 nm), which offers the highest absorption cross sections and fluorescence quantum yields, thus providing the most sensitive detection. Since other atmospheric species also exhibit strong absorption features in this UV region (e.g. the Hartley band of $O_3$ and the γ system of NO), techniques using broadband excitation sources (e.g. the lamp in the PF analyser used in this work) are susceptible to spectral interferences, thereby reducing their sensitivity and selectivity to $SO_2$. These interferences can be somewhat overcome by the use of bandpass filters to control the excitation wavelength, even more powerful however is the ability to resolve spectral features

of $SO_2$ by the use of a narrow band light source such as a laser. The combination of such a light source with the large absorption cross section of $SO_2$ at these wavelengths make LIF a prime candidate for achieving $SO_2$ detection with both very high sensitivity and specificity.

Remote sensing techniques (e.g. differential optical absorption spectroscopy (DOAS) and UV camera spectral imaging) primarily target the weaker $\tilde{B}$ band (~300-320 nm), which enables the utilisation of commercial UV cameras and detectors and coincides with less interference from other species but also corresponds to smaller $SO_2$ absorption cross sections than the $\tilde{C}$ band. Such remote detection techniques are, in general, not sufficiently sensitive to detect the low levels of $SO_2$ seen in clean marine air due to the smaller absorption cross sections at 300-320 nm and because they rely on absorption rather fluorescence detection."

Line 90: Reference Figure S4.

We have added a reference to Figure S4 on line 122 instead, as we deemed this a more appropriate position.

Line 183: "was" should be "were".

We thank the referee for this comment and have amended the text accordingly.

Figure 10: Personally, I think it would be neater to have the gradient and $R^2$ presented inside the plot like in Figure 8. Also, why is there no uncertainty on the CIMS-$SO_2$:$CO_2$ gradient?

We have added the gradient and $R^2$ values inside the plot for Fig. 10 and have also amended the similar regression plots in the SI. The lack of an uncertainty for the CIMS-$SO_2$:$CO_2$ gradient was primarily due to difficulties in accurately estimating an uncertainty for the CIMS. Based on the reviewers comments we realise that this is a significant omission from the work and have therefore estimated a CIMS uncertainty of 102 % based mainly on the fitting of the $SO_2$ peak. However, due to a large difference in instrument uncertainty between the CIMS $SO_2$ and $CO_2$ data, this effectively down-weights the CIMS data to the point that the fitted slope becomes poorly constrained and biased toward zero. Therefore, an uncertainty on the CIMS-$SO_2$:$CO_2$ gradient has been estimated from an ordinary least squares (OLS) fit and added to Fig. 10. For consistency, the regression plots in the SI have also been fitted via the OLS method.

Figure 11: ppb used in y axis label whereas ppbv mentioned in the text.

We have amended the text accordingly.

Line 404: Reference for the ACSIS project would be helpful – and possibly a Figure in the supplement of where it was flying.

We have added a reference to the "Data supporting the ACSIS programme" publication by Archibald et al. (2025) on line 437, which includes a flight track plot of ACSIS-7.

---

## Author Comment (AC3)

**Response to Referee 2**

We thank the referee for their positive remarks regarding this work particularly the characterisation of the LIF instrument and measurement protocols. We also feel that their detailed comments, particularly regarding the altitude plots, have enabled us to improve the manuscript, for which we are very grateful. Our responses to the referee's specific comments are as follows:

Line 174: "and to capture instrument drift": Did the authors notice any consistent drift for the LIF instrument over the course of a given flight? If so, what is the possible cause?

We have added the following text to explain the cause of the LIF instrumental drift:

"The observed drift resulted from variations in laser linewidth caused by external temperature fluctuations within the aircraft cabin, which directly impacts instrument sensitivity."

Line 181: "As a result of inconsistencies in the laser linewidth": How was this determined that the laser linewidth was the cause of variations in the sensitivity as opposed to errors in the calibration system, etc.? There is a lot of overlap in the error bars in Figs S1-S3, so it doesn't appear that the sensitivities are statistically different.

We determined the laser linewidth to be the cause of variations in sensitivity due to corresponding reference cell signal changes. We agree that the sensitivities are not statistically different and so have replaced the text "As a result of inconsistencies in the laser linewidth, the sensitivities were seen to vary slightly during the course of a flight" with:

"As the sensitivities during the course of a flight largely agree within errors"

Line 242: "larger interfering peak": What are some possible compounds that would correspond to the interfering peak in Figure 4(a) at m/z 190.899311? After seeing that figure, it is understandable why obtaining $SO_2$ from that m/z is so challenging.

We have added the following text to suggest a possible compound that would correspond to the interfering peak in Figure 4(a) at m/z 190.899311.

"likely an isotope of nitric acid"

Figure 8: I assume these are York regression fits being reported? Also, could the authors report the error on the slope and intercept for the fits (particularly Fig 8a)?

York regression was not used in Fig. 8 as we were unable to accurately quantify the large uncertainty on the CIMS observations. As the CIMS measurement uncertainty is significantly larger than the LIF, for the reasons explained in the paper, we felt a standard OLS was sufficient. In responding to the reviewers comments we have estimated a CIMS uncertainty of 102 % and attempted a York regression for this plot. However, due to the large difference in instrument uncertainty associated with the I⁻CIMS cps data compared with the LIF, this effectively down-weights the CIMS data to the point that the fitted slope becomes poorly constrained and biased toward zero. As the primary aim of these plots is to a) obtain a suitable gradient linking the CIMS cps data to the LIF mixing ratios for estimating the CIMS mixing ratios, and b) compare how well the PF and LIF mixing ratios agree, we feel a simple ordinary least squares (OLS) regression method is adequate and has been used for both plots for consistency. We have stated the fit technique in the caption of Fig. 8 and have added the errors on the slopes and intercepts from the OLS fits, as requested.

Lines 401-404: A comparison is made between $SO_2$ LIF data between the ACRUISE and ACSIS-7 campaigns, but no further analysis is done other than just showing the vertical profile of $SO_2$ in Fig S9. The authors should either develop this analysis further or consider cutting ACSIS-7 and Fig S9 from the manuscript.

We thank the referee for this comment and appreciate their perspective. While an in-depth analysis of $SO_2$ vertical profiles is outside the scope of this paper, we believe that including these ACSIS-7 data provides a valuable contribution to demonstrating the capabilities of the LIF instrument and for presenting rare observations of $SO_2$ vertical structure in the remote marine environment. We have therefore retained the reference to ACSIS-7 and Fig. S9 (as supported by Referee 1) and have added the following text comparing our observations with the only other reported vertical $SO_2$ data over a similar region in the North Atlantic, to our knowledge. However, if the editor agrees we should remove this material from the paper we will.

"It is evident that the marine environment sampled during ACSIS-7 was cleaner at the sea surface, likely reflecting the reduced influence of ship emissions compared to ACRUISE-3. $SO_2$ mixing ratios, however, become increasingly comparable to those measured during ACRUISE-3 at higher altitudes and display a similar decreasing trend with altitude up to 2000 m due to vertical mixing and diffusion from marine $SO_2$ sources. The ACSIS-7 profile exhibits a notable mixing ratio inversion at 2000 m, coincident with the marine boundary layer height.

While reported observations of $SO_2$ altitude profiles over the ocean are scarce, relevant comparisons can be drawn from measurements in the Atlantic between 30N – 54N during the AtoM campaigns (Bian et al., 2024). Considering differences in sampling strategies, time of year and location etc., the AToM profiles are consistent with our observations, showing $SO_2$

mixing ratios of a similar order of magnitude, albeit slightly lower, and a comparable decreasing trend with altitude (up to 3.8 km)."

Figure S10: Why is this figure included if not mentioned in the main text?

We thank the referee for this comment and have removed Figure S10 from the SI.

Figure 4 was hard to read since it appeared more pixelated than other figures.

We thank the referee for this comment and have improved the resolution of Figure 4.